



# Reviews and Syntheses: Spatial and temporal patterns in metabolic
# fluxes inform potential for seagrass to locally mitigate ocean
# acidification
Kristy J. Kroeker[1*], Tye L. Kindinger[1], Heidi K. Hirsh[2], Melissa Ward[3], Tessa M. Hill[3,4],
Brittany M. Jellison[3,6], David A. Koweek[5], Sarah Lummis[1], Emily B. Rivest[3,7], George G.
Waldbusser[8], Brian Gaylord[3,6]
[1]Department of Ecology and Evolutionary Biology, University of California Santa Cruz, Santa
Cruz, CA, USA
[2]Department of Earth System Science, Stanford University, Stanford, CA, USA
[3]Bodega Marine Laboratory, University of California Davis, Bodega Bay, CA, USA
[4]Department of Earth and Planetary Sciences, University of California Davis, Davis, CA, USA
[5]Department of Global Ecology, Carnegie Institution for Science, Stanford, CA. Present address:
Ocean Visions, Inc. 225 Baker Street, Atlanta, GA, USA
[6]Department of Evolution and Ecology, University of California Davis, Davis, CA, USA
[7]Department of Biological Sciences, Virginia Institute of Marine Science, William & Mary,
Gloucester Point, VA, USA
[8]College of Earth, Ocean and Atmospheric Sciences, Oregon State University, Corvallis, OR,
USA
*correspondence to: Kristy J. Kroeker (kkroeker@ucsc.edu)



**Abstract:** As global change continues to progress, there is a growing interest in assessing any
local levers that could be used to manage the social and ecological impacts of rising $CO_2$
concentrations. While habitat conservation and restoration have been widely recognized for their
role in carbon storage and sequestration at a global scale, the potential for managers to use
vegetated habitats to mitigate $CO_2$ concentrations at local scales in marine ecosystems facing the
accelerating threat of ocean acidification (OA) has only recently garnered attention. Early studies
have shown that submerged aquatic vegetation, such as seagrass beds, can locally draw down
$CO_2$ and raise seawater pH in the water column through photosynthesis, but empirical studies of
local OA mitigation are still quite limited. Here, we leverage the extensive body of literature on
seagrass community metabolism to highlight key considerations for local OA management
through seagrass conservation or restoration. In particular, we synthesize the results from 62
studies reporting *in situ* rates of seagrass gross primary productivity, respiration, and/or net
community productivity to highlight spatial and temporal variability in carbon fluxes. We
illustrate that daytime net community production is positive overall, and similar across seasons
and geographies. Full-day net community production rates, which illustrate the potential
cumulative effect of seagrass beds on seawater biogeochemistry integrated over day and night,
were also positive overall, but were higher in summer months in both tropical and temperate
ecosystems. Although our analyses suggest seagrass meadows are generally autotrophic, the
modeled effects on seawater pH are relatively small in magnitude. In addition, we illustrate that
periods when full-day net community production is highest could be associated with lower
nighttime pH and increased diurnal variability in seawater $p$$CO_2$/pH. Finally, we highlight
important areas for future research to inform the next steps for assessing the utility of this
approach for management.








## 1 Introduction

As carbon dioxide ($CO_2$) emissions continue to rise, there is an intense interest among managers

and decision makers in developing local strategies to minimize the social and ecological costs of

global change. While habitat restoration or conservation has been recognized for its utility in

carbon storage and sequestration at a global scale in both terrestrial and aquatic systems

(Canadell and Raupach 2008, Duarte et al. 2010, Agrawal et al. 2011, Mcleod et al. 2011, Nahlik

and Fennessy 2016), the potential for using vegetated habitat to mitigate $CO_2$ concentrations in

the surrounding environment at very local scales has received far less attention. In marine

ecosystems, however, increasing $CO_2$ concentrations cause ocean acidification (OA), which

poses a threat to species and ecosystems worldwide (Kroeker et al. 2010; 2013). Submerged

aquatic vegetation can reduce $CO_2$ concentrations in the water column through photosynthesis at

a local scale (hereafter termed *local OA mitigation*) (Hendriks et al. 2014). Thus, conservation,

restoration, and purposeful culturing of submerged aquatic vegetation have emerged as some of

the few potential strategies for local OA mitigation for managers, providing possible benefits to

other vulnerable species associated with these habitats (Washington State Blue Ribbon Panel on

Ocean Acidification 2012, Chan et al. 2016, Nielsen et al. 2018).


Because both carbon sequestration and local OA mitigation will depend on the productivity of

the vegetation, temporal variability in productivity may be especially important for managers

considering local OA mitigation strategies. Carbon storage and sequestration in marine





ecosystems is largely considered to be a cumulative process, whereas photosynthesis-based, local
OA mitigation in the water column will necessarily be more intermittent or transient in time. In
particular, variability in potential local OA mitigation can be expected on several different
temporal scales. First, daytime and nighttime patterns in photosynthesis and respiration can cause
substantial diurnal variability in seawater pH and saturation state (Hendriks et al. 2014, Pacella et
al. 2018), such that potential local OA mitigation will vary on hourly timescales with daylight.
Moreover, tidal cycles and local hydrodynamics may alter the impact these processes have on
ambient water chemistry on hourly timescales as well (Cyronak et al. 2018, Koweek et al. 2018).
Local hydrodynamics could significantly influence the time that a water mass experiences
chemical alteration by a seagrass meadow, as well as the water depth through which light must
penetrate to reach the seagrass canopy and the volume of water that must be modified, and
therefore the magnitude of the change. On a longer timescale, seasonal patterns in temperature
and light can also cause substantial seasonal variability in the biomass and productivity of
submerged aquatic vegetation (Maher and Eyre 2011, Clavier et al. 2014, Ricart et al. 2021).
Thus, the degree of any potential local OA mitigation may vary on seasonal or monthly
timescales as well (Manzello et al. 2011, Unsworth et al. 2012, Cryonak et al. 2018, Saderne et
al. 2019). This potential for temporal variability in photosynthesis-based, local OA mitigation
has important implications for how the strategy might be used by managers, particularly with
respect to whether potential OA mitigation aligns with windows of vulnerability for sensitive
living resources or how variability is integrated through time by important species.

Seagrasses are productive marine macrophytes that are considered net autotrophic at the global
scale (Duarte et al. 2010), suggesting that gross primary production (GPP, defined here as the



rate of oxygen production) should exceed respiration (R, defined here as the rate of oxygen
consumption, determined using incubations or measurements in the dark) on average. Assessing
the utility of seagrass meadows for local OA mitigation, however, requires a better
understanding of the spatial and temporal variability in GPP, R, and net community production
(NCP, defined here as the GPP-R, or the net carbon flux, when both photosynthesis and
respiration are accounted for). For example, diurnal variability in seawater carbonate chemistry
could be more pronounced in geographies that support higher daytime GPP and nighttime R
(Duarte et al. 2010). In addition, seasonal variability in NCP could be more pronounced in
locations where there are larger fluctuations in light and other environmental drivers, such as
high latitude or temperate ecosystems.

Scientists have been quantifying NCP in seagrass meadows for decades (Odum 1956), and
insight regarding the potential for temporal and spatial variability in carbon fluxes can be gained
using this literature. In particular, understanding whether GPP, R, and NCP vary predictably
across time and in different geographies can provide important, first order information about
when and where local OA mitigation approaches might be effective, as well as insight into
challenges that could arise in particular localities. Although potential local OA mitigation is
caused by changes in the dissolved inorganic carbon (DIC) in seawater, which can be influenced
by several other important biological and physical factors (Koweek et al. 2018, James et al.
2020), the relationship between $O_2$ fluxes associated with seagrass metabolism and seawater DIC
is roughly proportional (i.e., if $O_2$ production goes up, DIC in seawater will go down). While
empirical studies of changes in seawater DIC are currently limited, the comprehensive literature

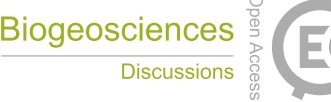

113 on seagrass community metabolism on $O_2$ fluxes can provide important spatial and temporal

114 context for managers interested in carbon fluxes and the potential for local OA mitigation.


116 Here, we synthesize published studies of seagrass metabolism to characterize the variability in

117 carbon fluxes associated with GPP, R, and NCP across seasons and geographies. In recognition

118 of the substantial temporal diel variability in carbon fluxes associated with daytime NCP and

119 nighttime respiration, as well as the uncertainty in our understanding of how this temporal

120 variability is integrated by vulnerable marine organisms associated with seagrass beds, we focus

121 on both hourly rates of NCP taken during peak daylight hours and full-day NCP. Hourly

122 measurements of NCP collected during peak daylight hours can provide insight into the

123 maximum elevation of seawater pH. Similarly, hourly measurements of respiration provide

124 insight into the potential maximum depression of nighttime pH. In contrast, measurements of

125 NCP taken over longer time periods or that incorporate the full 24 hour cycle (full-day NCP)

126 provide insight into the cumulative effect of seagrass on seawater chemistry. In particular, we

127 tested: (1) If seasonal variability is present in daytime and full-day carbon fluxes, (2) If the

128 temporal variation in carbon fluxes varies among tropical and temperate geographies, and (3)

129 How much of the residual variation in carbon fluxes not accounted for by seasons or geography

130 can be attributed to variation in temperature and aboveground biomass of the seagrass

131 assemblage. To connect the metabolic measurements to seawater chemistry, we model potential

132 changes in bulk seawater pH based on the estimated carbon fluxes given variation in seawater

133 residence time and water depth.


135 **2 Methods**



We conducted a literature search for *in situ* measurements of seagrass community metabolism
using the Web of Science. Search terms included *[seagrass* OR *eelgrass* OR *submerged aquatic*
*vegetation]* AND [*metabol** OR *carbon/oxygen fluxes* OR *community prod* community resp**
OR *benthic incubation chambers* OR *primary prod** OR *carbon* chemistry* OR *pH]*. For each
paper, we then searched the literature cited for more applicable studies, as well as any papers
listed in the Web of Science that cited the study in question. In addition, we searched the datasets
used by Duarte et al. (2010) and Unsworth et al. (2012). Studies were included when either $O_2$ or
carbon fluxes of a seagrass-associated community were measured *in situ.* This included studies
using a variety of methods, including incubation chambers, eddy correlation techniques, mass
balance estimates, and isotope enrichment, among others. Studies were included that were
published prior to January 1, 2020.

Within a single study, regardless of the methods used, each deployment/set of measurements was
included as a data point in the synthesis when deployments/measurements were repeated across
different locations, months, or species. We collected measurements of GPP, R or NCP from each
study using data reported in the text, tables, or graphs using software (Graph Click or Data
Thief), or provided by the authors by request. In addition, we recorded information on the
photosynthetic quotient (PQ) and respiratory quotient (RQ) values used to convert from $O_2$ to
carbon, as well as other metadata associated with the study (e.g., species, location, temperature,
month the study was conducted, etc.). We classified each study as either tropical or temperate
based on the designation in the primary study and then classified the metabolic measurements as
either (a) hourly rates or (b) daily rates. This classification was defined by the reporting within
the studies (i.e., the primary authors either reported hourly or daily rates), but the difference in





reporting was ostensibly due to differences in the length of the deployment used to measure
metabolism (e.g., <4 hour deployment = an hourly rate, ~12-24 hour deployment = a daily rate).
The shorter "hourly" deployments were usually taken during peak daylight hours, which we used
to infer the potential for any daytime local OA mitigation. In contrast, we use the daily rates to
infer the cumulative, full-day local OA mitigation potential of seagrass. It is important to note
that positive daily NCP (used to infer the full-day local OA mitigation potential) can still
encompass marked diel or diurnal variability in carbon fluxes that could prove deleterious to
seagrass associated species during transient periods of low pH.

Although no studies included here measured changes in seawater DIC directly, several studies (N
= 17) converted metabolism measurements based on oxygen consumption and production to
units of carbon. For studies that only reported metabolism in units of oxygen, we converted the
reported GPP, R, and NCP measurements to carbon using a PQ or RQ of 1 (Duarte et al. 2010).
We then converted all measurements to the same scale: either *mmol C/m²/hour* or *mmol*
*C/m²/day*. Positive NCP values represent net autotrophy and carbon fluxes *from* the water
column *to* the seagrass tissue, and negative NCP values represent net heterotrophy and carbon
fluxes *from* the seagrass tissue *to* the water column.

To assess differences in estimates based on the methods used to measure metabolism, we plotted
the carbon fluxes as a function of study type. Based on these plots (Fig. S1), we decided to
perform separate analyses for studies that used the "mass balance" approach (Odum 1956) versus
other methods (e.g., incubations, eddy correlations). The studies using the mass balance
approach often do not differentiate between water column and benthic productivity, and as such,





displayed a much higher range and magnitude of responses than those measured by other
methods (see *Methodological Analyses* in *Results* below).

To assess the drivers of variability in carbon fluxes (hourly and daily rates of GPP, R and NCP),
we first assessed the collinearity in the primary drivers of interest: seawater temperature and
aboveground biomass for those studies reporting both variables. Because temperature and
aboveground biomass are correlated (temperature × geography: $P = 0.011$, Fig S2), we decided
to focus first on the effect of season, using *month* as a predictor variable, which ostensibly
encompasses some of the variability in both temperature and aboveground biomass (Figs S3 and
S4). Furthermore, the use of month as a predictor variable allowed us to include the maximum
number of studies in the analysis, since not all studies reported temperature and aboveground
biomass. To standardize months to seasons across the hemispheres, we used the numerical
notation for months in the northern hemisphere (i.e., January = 1, etc.). For the southern
hemisphere, we subtracted 6 from the numerical notation and used the absolute value. In
addition, we tested for differences between seagrass communities in temperate and tropical
geographies based on the hypothesis that seagrass meadows in temperate geographies have
greater seasonality in light, temperature, and aboveground biomass, and thus, should have a more
pronounced seasonal fluctuation (Fig S3 and S4).

We then tested for effects of temperature and aboveground biomass on the residual variation of
the monthly models. Specifically, we first fit mixed-effects models of both hourly and daily rates
of GPP, R, and NCP using maximum likelihood with *geography* as a categorical factor (tropical
vs. temperate) and a linear and quadratic term for *month* as fixed effects, as well as the two-way





interactions between *geography* and each term for *month*. We also included *replicate* nested in
*study* as a random effect to account for non-independence arising from the inclusion of repeated
measures from the same sites over time and measurements from multiple sites within a single
study:
$$metabolism \sim month \ + \ month^2 \ + \ geography \ + \ month \ \times \ geography$$
$$+ \ month^2 \ \times \ geography \ + \ 1|study(replicate)$$

We then used backwards model selection to determine the significance of fixed effects based on
likelihood ratio tests. Final models were fit using restricted maximum likelihood to calculate
model estimates. We then performed two separate analyses using (1) environmental temperature
and (2) aboveground biomass to assess any remaining variability in the residuals from the
seasonal models. First, using just the subset of studies that either reported temperature or
biomass, we fit the final seasonal models again using restricted maximum likelihood to obtain
the conditioned residuals. Then, using these residuals, we fit linear models with *geography* as a
categorical factor, plus a linear term for either *temperature* or aboveground *biomass* as well as
the interaction between *geography* and *temperature/biomass* as fixed effects. We used
backwards model selection, comparing nested models with a series of ANOVAs. Finally, we
tested for net autotrophy (NCP>0) using a one-tailed t-test. All analyses were performed in R
(version 3.6.2) (R Core Team 2019)  with the following packages as needed: nlme (version
3.1.145) (Pinheiro et al. 2020), broom (version 0.7.3) (Robinson et al. 2020), and broom.mixed
(version 0.2.6) (Bolker and Robinson 2020).





To illustrate how water depth and residence time may mediate the effects of the carbon fluxes
associated with seagrass communities on bulk seawater pH for potential local OA mitigation, we
applied the range of hourly net carbon fluxes (NCP) covered in our synthesis to a simplified,
steady state box model developed by Koweek et al. (2018). We use the hourly rate rather than
the full-day rate because we recognize that the effects of seagrass on seawater carbonate
chemistry will be intermittent and fluctuate over the daylight hours. We modeled the change in
dissolved inorganic carbon as a function of NCP as
$$\Delta DIC = \frac{L}{\bar{u}\rho h} \times NCP$$

where $L$ = the box length (m), $\bar{u}$ = mean water velocity (m s$^{-1}$), $\rho$ = is the seawater density (kg
m$^3$), and $h$ = water depth (m),. Because of the familiarity among managers and decision makers
with seawater pH, we then converted the delta DIC to pH, assuming a relevant, temperate coastal
ocean condition (e.g., total alkalinity = 2300 $\mu$mol/kg, temperature = 15°C, and salinity = 35
ppm). We then plotted the change in pH as a function of hourly daytime carbon fluxes (i.e.,
hourly NCP) for two different water depths (0.5 and 2m) and three different water residence
times ($L/\underline{u}$ = 15 minutes, 60 minutes, and 4 hours) at each water depth. We selected four hours
as the maximum duration for the model for two reasons: seagrass beds are rarely extensive
enough for water to remain over seagrass for more than a few hours, and longer residence times
would tend to overlap with lower-light conditions when the hourly NCP does not apply.

**3 Results**
*3.1 Description of the database*
Using our search criteria, we identified 62 published papers (spanning 1956 to 2020) that
reported *in situ* rates of seagrass community metabolism (Table S1). None of the studies directly





measured changes in DIC, but 17 of the 62 studies reported metabolic measurements in units of
carbon. The complete set of studies spanned temperate and tropical ecosystems. The inclusion of
36 temperate studies and an additional 15 tropical studies significantly expanded the scope of
inference beyond previous reviews (Duarte et al. 2010; Unsworth et al. 2012). Many studies
occurred in the Western Atlantic and Mediterranean (Fig. 1), and there were no studies from the
North Pacific. Most studies measured seagrass metabolism during the spring and summer
months, while fewer studies measured the metabolism in fall and winter conditions (Fig. 1).
Environmental temperature was highest during late summer/early fall months and was higher
overall in tropical biomes (Fig S3). Aboveground biomass was highest during summer months
and higher in the temperate geographies (Fig S4).

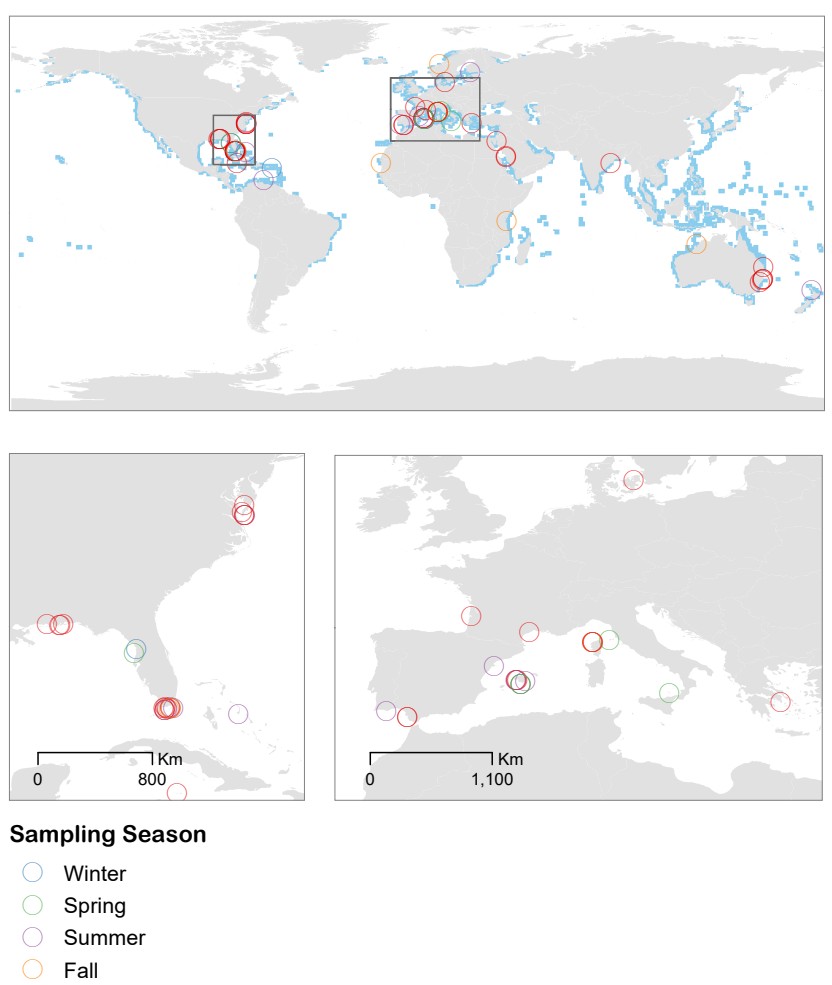

**Figure 1.** Studies included in the analyses span temperate to tropical ecosystems, with many studies occurring in the Western Atlantic and Mediterranean. Most studies measured seagrass metabolism during the spring and summer months, while fewer studies measured the metabolism in fall and winter conditions. Blue pixels represent the distribution of seagrasses from UNEP-WCMC and Short (2018).



*3.2 Methodological analyses*
Our results illustrate greater variability in the ranges of response observed using the "mass
balance" method, which extend in magnitude beyond those observed using other methods for
measuring both GPP and R (Fig. S1 a-f). This greater variability does not appear to be driven by
timing of the measurements as the "mass balance" method produced metabolic measurements of
higher variability or magnitude across seasons (Fig S1 g-l). Measurements taken using
incubations, eddy correlation methods, and water column measurements of pH using *in situ*
sensors and an acoustic doppler velocimeter (or other instrument capable of measuring flow) are
generally of similar magnitude and variation.

*3.3 Spatial and temporal patterns in carbon fluxes*
*3.3.1. Daytime carbon fluxes*
Measurements of hourly carbon fluxes (N=83 for NCP), typically obtained from shorter duration
deployments conducted during peak sunlight hours, reveal differences in seasonal patterns of
GPP and respiration. Both GPP and R peak during summer months across both ecosystems (Fig
2a-b). Despite higher biomass in temperate systems during summer months (Fig S4), we do not
detect a statistical difference in the seasonal patterns among GPP in temperate and tropical
ecosystems (Table 1). This result is highly influenced by two studies in tropical geographies
(Morgan and Kitting 1984, Herbert and Fourqurean 2008); when these studies are not included,
summertime GPP is higher in temperate geographies than in tropical geographies (Fig S5).
Similarly, R peaks in summer months in both temperate and tropical ecosystems, and we detect a
sharper increase and a higher seasonal peak in R in temperate ecosystems (Fig 2b; Table 1). The
seasonal peaks in GPP and R effectively cancel each other out, resulting in no statistically





detectable difference in hourly NCP rates across seasons (Fig 2c). Although the net hourly
carbon flux associated with NCP does not vary seasonally, the mean net hourly carbon flux from
the seawater to the seagrass is positive (mean hourly NCP = 5.32 +/- 5.93 SD mmol C/m$^2$/hour),
indicating a net draw down of seawater DIC during peak daylight hours regardless of geography
(one-tailed t-test: $t_{82}$ = 8.18, P<0.001). Ninety-three percent of the 83 measurements were net
autotrophic.

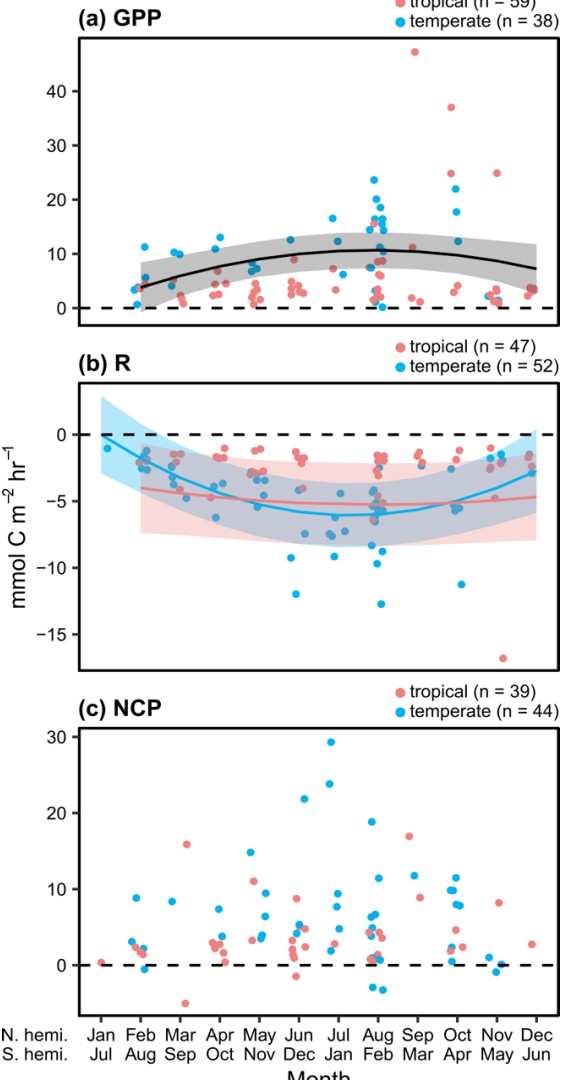

**Figure 2.** Hourly rates for C fluxes (mmol$^{-1}$m$^{2-1}$hour$^{-1}$) associated with seagrass communities as a function of time, with GPP measurements taken during peak sunlight hours (~10 am to 2 pm local time). Studies performed in temperate versus tropical ecosystems are illustrated by color (blue = temperate, red = tropical), and significance (p<0.05) is denoted by a fit line with a quadratic term and 95%CI. The colors of the lines denote significant differences between tropical and temperate systems in GPP and R, while a black line denotes a significant relationship but no difference between temperate and tropical geographies.





**Table 1.** Statistics for linear mixed effects models of hourly rates of carbon fluxes. Terms in grey
were removed from the final model using backward model selection.

| Response | Outliers removed | Fixed effect | df | L-ratio | p-value |
|---|---|---|---|---|---|
| GPP (mmol C/m²/hr) | 0 | month | | | |
| | | month² | 5 | 6.76905 | **0.0093** |
| | | geography | 6 | 0.73356 | 0.3917 |
| | | month × geography | 7 | 0.00066 | 0.9795 |
| | | month² × geography | 8 | 0.32073 | 0.5712 |
| | 4 | month | | | |
| | | month² | 7 | 13.7499 | **0.0002** |
| | | geography | | | |
| | | month × geography | 8 | 4.60594 | **0.0319** |
| | | month² × geography | 8 | 3.5689 | 0.0589 |
| R (mmol C/m²/hr) | 0 | month | | | |
| | | month² | | | |
| | | geography | | | |
| | | month × geography | 8 | 5.98346 | **0.0144** |
| | | month² × geography | 8 | 5.72857 | **0.0167** |
| | 1 | month | | | |
| | | month² | | | |
| | | geography | | | |
| | | month × geography | 8 | 9.23633 | **0.0024** |
| | | month² × geography | 8 | 8.73371 | **0.0031** |
| NCP (mmol C/m²/hr) | 0 | month | 4 | 0.04176 | 0.8381 |
| | | month² | 5 | 2.65064 | 0.1035 |
| | | geography | 6 | 0.04703 | 0.8283 |
| | | month × geography | 7 | 3.41018 | 0.0648 |
| | | month² × geography | 8 | 1.09949 | 0.2944 |








*3.3.2 Full-day carbon fluxes*
We found 164 measurements/deployments that reported full-day NCP using methods that span a
wider range of photoperiods and environmental conditions, and thus provide insight into the
potential for full-day local OA mitigation. Based on the accompanying daily rates of GPP and R,
there is evidence of a seasonal cycle in carbon fluxes to and from the water column associated
with seagrass metabolism (Fig. 3a-b). The seasonal fluctuation differed statistically between
temperate and tropical geographies, with a sharper slope in the seasonal fluctuation among the
tropical studies (Fig 2a). We did not detect a difference in R between geographies. In general, the
seasonal fluctuation in GPP exceeds the seasonal fluctuation in respiration in both geographies,
resulting in higher daily net carbon flux from the seawater to the seagrass associated with NCP in
summer months (Fig. 3c). The seasonal fluctuation in NCP was greater among the tropical
studies than the temperate studies (Table 2). The mean NCP for tropical geographies was 62.5
(+/- 62.4 SD) mmol C/m$^2$/day, with 84% of the 77 total reported measurements being
autotrophic. The mean NCP for temperate geographies was 28.8 (+/- 79.0) mmol C/m$^2$/day, with
68% of the 187 total reported measurements being autotrophic. Overall, the seagrass meadows in
both geographies were net autotrophic (*one-tailed t-test:* tropical $t_{76}$=8.78, P<0.001; temperate
$t_{186}$ = 4.98, P<0.001). Despite these overall trends, there are several individual studies that
reported net heterotrophy and net carbon fluxes to the water column, even during summer
months.

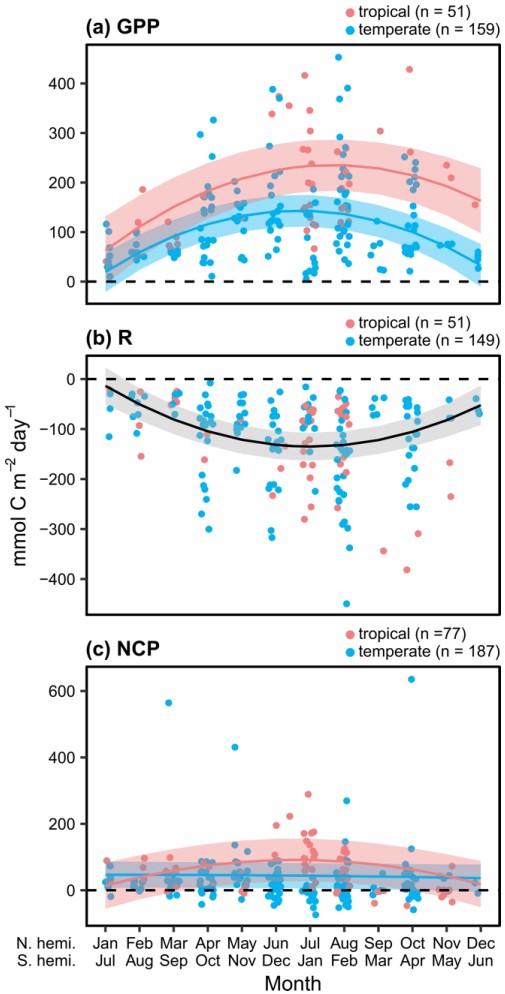

**Figure 3.** Daily rates for C fluxes (mmol$^{-1}$m$^{2-1}$day$^{-1}$) associated with seagrass communities as a function of time. Studies performed in temperate versus tropical ecosystems are illustrated by color (blue = temperate, red = tropical), and significance ($p<0.05$) is denoted by a fit line with a quadratic term and 95%CI. The colors of the lines denote significant differences between tropical and temperate systems.



**Table 2.** Statistics for linear mixed effects models of daily rates of carbon fluxes. Terms in grey
were removed from the final model using backward model selection.

| Response | Fixed effect | df | L-ratio | p-value |
|---|---|---|---|---|
| GPP (mmol C/m²/day) | month | | | |
| | **month²** | 7 | 54.5932 | **<0.0001** |
| | geography | | | |
| | **month × geography** | 7 | 4.28394 | **0.0385** |
| | month² × geography | 8 | 0.05199 | 0.8196 |
| R (mmol C/m²/day) | month | | | |
| | **month²** | 5 | 40.5399 | **<0.0001** |
| | geography | 6 | 2.7621 | 0.0965 |
| | month × geography | 7 | 2.2282 | 0.1355 |
| | month² × geography | 8 | 2.04071 | 0.1531 |
| NCP (mmol C/m²/day) | month | | | |
| | month² | | | |
| | geography | | | |
| | **month × geography** | 8 | 16.9111 | **<0.0001** |
| | **month² × geography** | 8 | 15.1501 | **<0.0001** |





*3.4 Drivers of chemical variability*
Within seasons, there is still marked variation in GPP and respiration (Fig. 2-3). Using the subset
of studies that report environmental temperature (N = 28), we found that temperature did not
explain the residual variability in any metric besides hourly GPP (Fig. 4; Table 3), suggesting the
seasonal models may generally account for hypothesized temperature effects. As noted,
temperature explained some of the residual variability from the seasonal models of hourly GPP,
with the effect differing among tropical and temperate geographies (Fig S6; Hourly GPP
*Geography x Temperature*: $F_{42}$ = 10.83, P = 0.001). Among studies reporting aboveground
biomass (N=23), biomass explains some of the residual variability in daily NCP, although the
effect depends on geography as well (Fig 5; Table 3). Aboveground biomass also explains some
of the residual variability in the seasonal models of hourly GPP, respiration, and NCP, and the
effect of biomass on hourly GPP also depended on the geography (Fig. S7; Table 3).

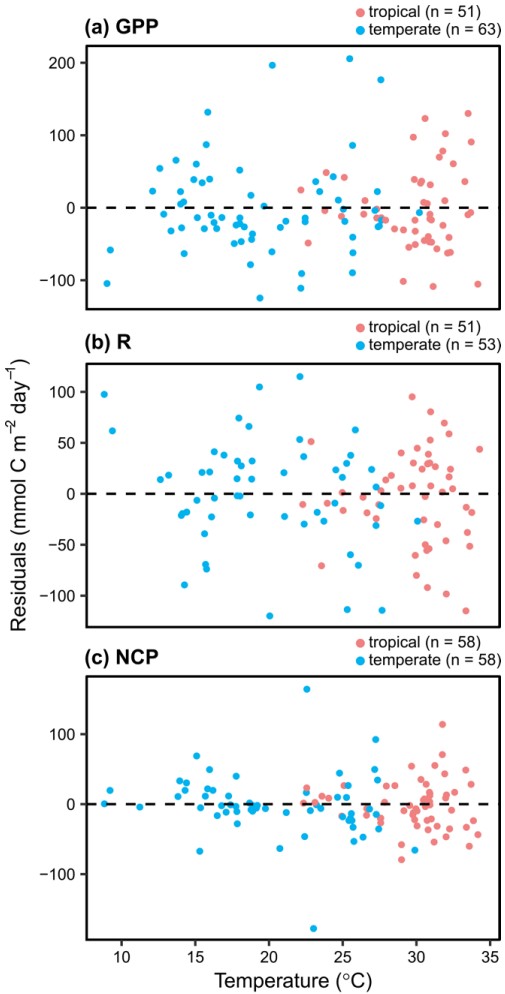

**Figure 4.** Conditioned residuals of the daily rates of C fluxes ($mmol^{-1}m^{2-1}day^{-1}$) from a seasonal

model as a function of temperature. Studies performed in temperate versus tropical ecosystems

are illustrated by color (blue = temperate, red = tropical). None of the relationships are

statistically significant.

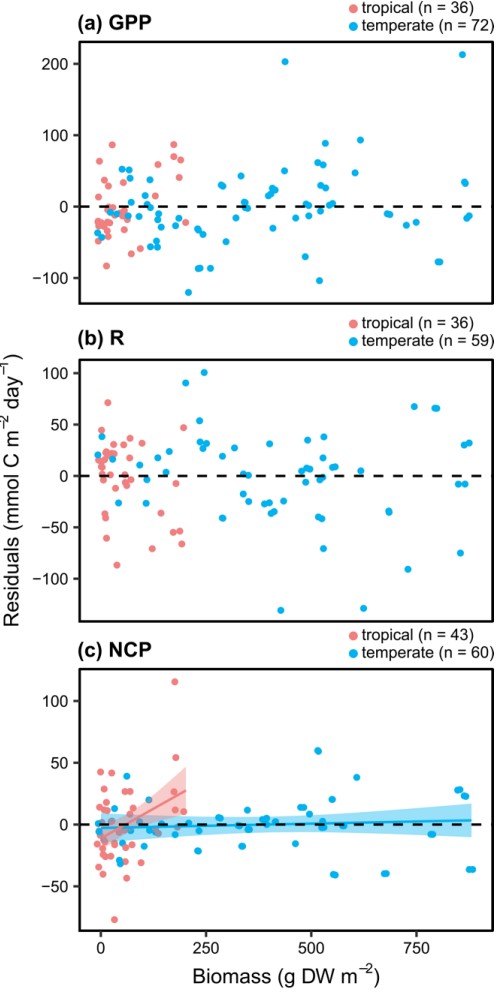

**Figure 5.** Conditioned residuals of the daily rates for seagrass C fluxes (mmol$^{-1}$m$^{2-1}$day$^{-1}$) from a

seasonal model as a function of aboveground biomass measured in the field during metabolism

measurements. Studies performed in temperate versus tropical ecosystems are illustrated by color

(blue = temperate, red = tropical). Significance is denoted by a fit line and 95% CI.








**Table 3.** Statistics for mixed effects models of the residuals of the hourly metabolic rates as a
function of biomass or temperature and geography and the interactions.

| Response | Fixed effect | df$_{res}$ | SS$_{res}$ | SS | F | p-value |
|---|---|---|---|---|---|---|
| GPP (mmol C/m²/hr) | biomass | | | | | |
| | geography | | | | | |
| | **biomass × geography** | 33 | 2806.1 | -1081.7 | 20.074 | **<0.0001** |
| R (mmol C/m²/hr) | biomass | 30 | 175.9 | -20.68 | 3.864 | 0.059 |
| | geography | 30 | 161.9 | -6.681 | 1.248 | 0.273 |
| | biomass × geography | 29 | 155.3 | -1.505 | 0.274 | 0.605 |
| NCP (mmol C/m²/hr) | **biomass** | 38 | 1640.2 | -186.1 | 4.736 | **0.036** |
| | **geography** | 38 | 1721.4 | -267.4 | 6.803 | **0.013** |
| | biomass × geography | 37 | 1454.1 | -140.3 | 3.844 | 0.058 |
| GPP (mmol C/m²/hr) | temp | | | | | |
| | geography | | | | | |
| | **temp × geography** | 42 | 375.4 | -71.84 | 9.704 | **0.003** |
| R (mmol C/m²/hr) | temp | 61 | 221.7 | -0.930 | 0.253 | 0.617 |
| | geography | 60 | 220.8 | -0.104 | 0.028 | 0.868 |
| | temp × geography | 59 | 220.7 | -0.004 | 0.001 | 0.976 |
| NCP (mmol C/m²/hr) | temp | 46 | 1453.0 | -41.82 | 1.334 | 0.254 |
| | **geography** | 47 | 2141.0 | -688.0 | 21.78 | **<0.0001** |
| | temp × geography | 45 | 1411.2 | -12.30 | 0.387 | 0.537 |












*3.5 Potential OA amelioration*


The steady state box model illustrates that the largest potential change in seawater pH occurs
when NCP is highest and the water depth and residence time are lowest (Fig. 6). Using the mean
hourly NCP from our analysis (~5.32 mmol C/m$^2$/hour; Fig. 2C), the potential change in
seawater pH in a seagrass meadow that meets the assumptions of the box model (e.g., $\Delta O_2$:$\Delta DIC$
= 1) in a low flow environment at low tide (i.e., 0.5m water depth) ranges from 0.006 – 0.085 pH
units for a residence time from 15 minutes to 4 hours. At the modeled higher tide (i.e., 2m water
depth), the potential changes in seawater pH for the same NCP range from 0.001-0.022 pH units
for the same residence times (15 minutes to 4 hours).




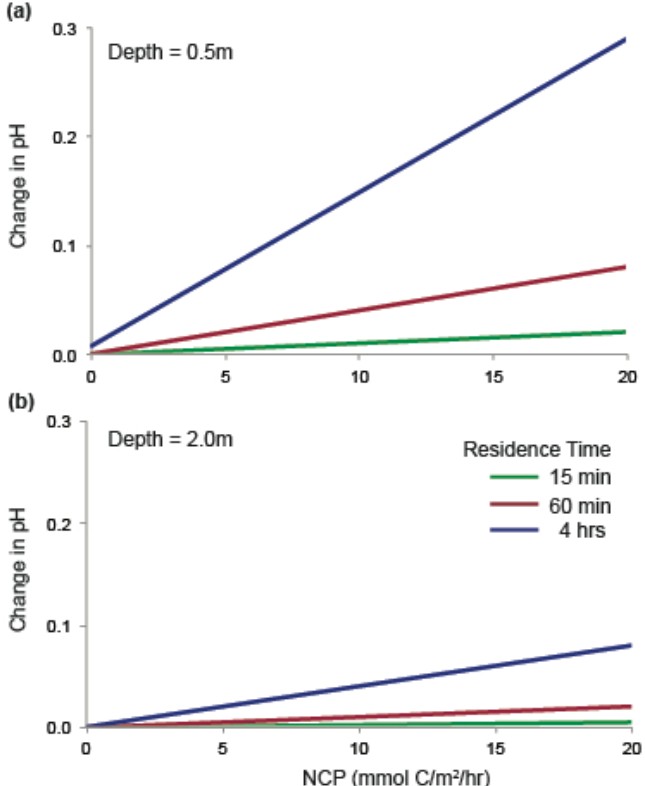

**Figure 6.** Results from a steady state box model illustrating the change in seawater pH as a

function of NCP at (A) 0.5 m water depth and (B) 2 m water depth. The change in seawater pH is

illustrated for a series of different residence times (L/u). Both panels are modeled at the same

temperature (15°C), total alkalinity (2300 μmol/kg), and salinity (35 ppm), with an incoming

DIC of 2100 μmol/kg.





**4 Discussion**

*4.1 Spatial and temporal variability in seagrass metabolism*

Here, we report that the NCP of seagrass beds during daylight hours is positive and similar

across seasons and geographies (Fig 2C). This is ostensibly due to GPP generally exceeding R

during daylight and the seasonal fluctuations in hourly rates of GPP and R balancing each other

out. If $O_2$ fluxes translate proportionally to community drawdowns in $CO_2$, the assumption

underlying our box model, our results suggest that the maximum potential local OA mitigation

due to seagrass metabolism during daylight hours is similar across time and ecosystems, but

small in magnitude.

We also demonstrate that seagrass beds are generally net autotrophic over the length of the day

(based on daily NCP), and the magnitude of this full-day NCP is more pronounced during

summer months and in tropical geographies (Fig 3C). However, underlying the summertime

peak in full-day NCP is the potential for marked diurnal variability in pH. In particular, the

demonstrated summertime peak in hourly respiration rates could drive more pronounced

nighttime lows in pH/saturation state during the most pronounced windows for net autotrophy

(i.e., summer months with the highest daily NCP). However, diurnal fluctuations in seawater pH

associated with seagrass metabolism will also be influenced by hydrodynamics that are not

captured in our synthesis (Koweek et al. 2018), and recent field studies of estuarine eelgrass by

Ricart et al. (2021) demonstrated that sustained elevations in pH associated with seagrass

meadows were not restricted to daylight hours. Regardless, a critical next step for understanding

the utility of seagrass beds for climate adaptation is to determine how the demonstrated seasonal





variability and the potential diurnal variability in carbonate chemistry are integrated over time by
organisms, especially under conditions projected for coastal habitats in the future.

Given the positive correlations between GPP and temperature, we assume that most of these
measurements were taken when the environmental temperatures were not physiologically
stressful to the seagrass community. Thus, the positive relationship between environmental
temperature and GPP should not necessarily be viewed through the lens of temperature
exposures associated with future warming driven by climate change. Continued warming
associated with climate change could cause photosynthesis and respiration to decline at stressful
temperatures. The probability that the relationship between temperature and metabolic responses
will change with future warming is likely to differ geographically based on how close a
community is to its thermal limit and the scope for acclimation or adaptation. This may partially
explain the differences in the relationships between temperature and the residuals of the seasonal
model between geographies, with increasing temperature negatively related to the residuals from
tropical geographies and positively related to the residuals from temperate geographies. Many
tropical seagrass species are growing close to their photosynthetic and physiological optima (Lee
et al. 2007, Koch et al. 2012), and elevated temperatures in these geographies may be
detrimental, causing metabolism (in this case hourly GPP) to be less than expected based on
seasonal patterns. In contrast, the positive relationship between the residuals from the seasonal
model and temperature in temperate geographies indicate that there may be a stimulatory effect
before a thermal tolerance threshold is crossed and metabolism decreases.



The significant relationships between aboveground biomass and the residual variation in hourly
GPP and NCP indicate that aboveground biomass also plays an important role in carbon fluxes
beyond that which is already captured by any seasonal fluctuations in biomass. Although the
relationships between aboveground biomass and the residual variation in hourly GPP are
generally what would be expected (higher biomass = higher metabolic rates than expected based
on the seasonal model), the negative relationship between aboveground biomass and the residual
variation in hourly NCP is somewhat surprising. This relationship suggests that
deployments/measurements in seagrass beds with higher aboveground biomass generally had
lower hourly NCP than what is predicted by the seasonal model. This negative relationship may
be explained by self-shading in dense meadows, or it could be due to other organisms that
contribute to daytime respiration (e.g., heterotrophs) that are associated with the higher biomass
meadows due to its structural complexity or other habitat features, but are not accounted for in
the aboveground biomass measurements. Dedicated experiments may be able to determine the
mechanism for these findings; however, the positive relationships between aboveground biomass
and the residual variation in daily NCP suggests that, overall, higher aboveground biomass
generally increases production relative to respiration.

*4.2 Implications for local OA mitigation and management*
The results of our steady state box model analyses illustrate the potential scope for seagrass NCP
to influence seawater pH on an hourly basis (Fig. 6), with the change in pH being proportional to
NCP during daylight hours and R during nighttime hours. While the box model is useful in
making coarse estimates on what particular NCP values might correspond to in seawater pH, it is
important to note that it only represents a first step in translating the seagrass community



metabolism estimates to seawater biogeochemistry. This is in part because the ratio of NCP
based on carbon fixed and oxygen evolved in seagrass communities is likely to be quite variable.
Although the ratio between $O_2$ produced and carbon fixed by an individual seagrass is generally
assumed to be balanced (i.e., 1:1), the other processes that occur in a seagrass meadow, including
respiration from organisms living within the seagrass and carbonate production and dissolution,
also influence the dissolved inorganic carbon (DIC) concentration in the seawater. Current
empirical measurements of $NCP_{DIC}:NCP_{O2}$ in seagrass meadows range from 0.3 to 6.8 (Ziegler
and Benner 1998, Barrón et al. 2006), suggesting the effect of seagrass NCP on seawater pH
could be substantially more or less pronounced than illustrated here. Because of this variability
in the relationship between $O_2$ and DIC, care must be taken when interpreting the results from
the box model. A better understanding of the $NCP_{DIC}:NCP_{O2}$ in particular meadows will better
inform their potential for local OA mitigation. Finally, the utility of seagrass as a climate
mitigation tool will depend on the goal of the management, and in most cases, will require more
research. For example, if the goal of management is to prevent negative effects of ocean
acidification on oyster growth, then studies that quantify the sensitivity of oyster growth to the
variability in pH observed here are still required.

**5 Conclusions**
Few conservation or restoration efforts currently take into account the potential chemical
ecosystem services of seagrasses and other submerged aquatic vegetation. Here, we demonstrate
that daytime carbon fluxes associated with seagrass metabolism are likely to be similar across
seasons and geography, while the full-day carbon fluxes peak during summer months in both
tropical and temperate geographies. Integrating across seasons, seagrass meadows are net



autotrophic. However, our simplified model results suggest the daytime carbon fluxes reported
across the global ocean may translate to small changes in seawater pH. These seasonal patterns
largely capture the present-day effects of variability in temperature and aboveground biomass on
seagrass metabolism, but likely do not adequately model the effects of future warming as it
becomes physiologically stressful.

These results highlight challenges, as well as gaps in our understanding, that may impede the use
of seagrasses for sustained local OA mitigation. In particular, we demonstrate that while peak
daytime carbon fluxes are similar across seasons and geographies, nighttime respiration is
highest during summer months. Thus, although seagrass beds are generally net autotrophic,
nighttime respiration could reduce seawater pH during periods of greatest autotrophy. We
provide examples of how water depth and residence time can influence the effect of seagrass on
seawater pH, and we demonstrate that the overall magnitude of the effect is likely quite small.

This work has elucidated several gaps that need to be addressed by the scientific community. For
example, certain geographies, such as the North Pacific, are currently underrepresented in our
dataset. Thus, continued study of seagrass metabolism and its effects on seawater carbonate
chemistry are needed to expand our area of inference. In addition, studies are needed to constrain
the relationship between dissolved oxygen fluxes and DIC, and this relationship may be
important to elucidate at local scales to truly understand the potential for OA mitigation at a
given location. Perhaps most importantly, more information is needed to understand how
vulnerable organisms respond to the chemical variability highlighted in our study (Gimenez et al.
2018, Lowe et al. 2018), and in particular, how this variability is integrated through time. Despite



the considerations of geographic and temporal variability in carbon fluxes illustrated here, we
recognize that seagrass conservation and restoration may be important strategies for climate
adaptation for numerous other reasons, including carbon sequestration and habitat provisioning.

**6 Code Availability:** Code is available at
https://github.com/tyekindinger/SeagrassCommunityMetabolism

**7 Data Availability:** All data used in this analysis is publicly available via the published studies.

**8 Author Contributions:** Kroeker: conceptualization, funding acquisition, investigation,
methodology, writing – original draft preparation; Kindinger: formal analysis, visualization,
writing – review & editing; Hirsh: investigation, visualization, writing – review & editing; Ward:
investigation, writing – review & editing; Hill: conceptualization, funding acquisition, writing –
review & editing; Jellison, Koweek, Lummis, Rivest, Waldbusser: conceptualization, writing –
review & editing; Gaylord: conceptualization, funding acquisition, methodology, writing –
review & editing.

**9 Competing Interests:** The authors declare no competing interests

**10 Acknowledgments**
We would like to thank K. J. Nickols and Y. Takeshita for their helpful comments and
suggestions, which greatly improved the manuscript. This work was initiated by a working group
of seagrass and biogeochemistry experts, convened at Bodega Marine Laboratory with support



from California Sea Grant. The publication was prepared by K. J. Kroeker under NOAA Grant #
NA14OAR4170075, California Sea Grant College Program Project # R/HCME-03, through
NOAA'S National Sea Grant College Program, U.S. Dept. of Commerce. The statements,
findings, conclusions and recommendations are those of the author(s) and do not necessarily
reflect the views of California Sea Grant, NOAA or the U.S. Dept. of Commerce. In addition, K.
J. Kroeker and T. Kindinger received support from the David and Lucile Packard Foundation and
K. J. Kroeker received funding from the Alfred P. Sloan Foundation.

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
