# Peer review of "Reviews and Syntheses: Spatial and temporal patterns in metabolic"

_Biogeosciences, 2021_

## Referee Comment (RC1)

Kroeker et al report on a meta-analysis of seagrass ecosystem metabolism from a selection of recent studies. Major goals of this study were to identify variations in ecosystem metabolism due to 1) seasonality, 2) regional differences, and 3) biological/thermal drivers, and determine the extent to which seagrass ecosystem metabolism may help to mitigate coastal OA. Unfortunately, this study suffers on two major fronts. First, the literature considered in this meta-analysis is very incomplete, and for some reason only considered dissolved oxygen-based studies (omitting all studies estimating NCP from carbonate system measurements). Secondly, the very simplistic (bordering on sloppy) approach to carbonate chemical accounting is troubling, given that the key aim of the meta-analysis was to address possible OA effects of seagrass metabolism. Because of these concerns, I find that the conclusions of the study are not supported by the analysis presented, making this work unsuitable for publication at Biogeosciences. To appropriately address the key questions of this study will require a complete re-compilation and analysis of the dataset, with an eye towards more-appropriate carbonate chemical accounting.

Specific comments:

136: The authors raise two concerns regarding the existing literature, 1) that no data exist for the N Pacific, and 2) no studies exist using direct DIC measurements. For 1), the authors should see Tokoro et al (2014) and more recent work from the group. Regarding point 2), there are in fact quite a few studies which have relied directly on DIC or DIC+TA measurements to quantify NCP in seagrass meadows. At least one of these studies (Van Dam et al., 2019) was included in this meta-analysis, and it is not clear why the authors chose to omit these and other DIC-based metabolism estimates. Others including DIC measurements include, but are probably not limited to: Perez et al 2018; Eyre et al., 2011; Ribas-Ribas et al., 2011, Dollar et al., 1991; Turk et al., 2015

143-145: There is another major conceptual flaw here regarding the treatment of data collected using different methods. It is unclear what the authors intend NCP to represent. Is this only a consideration of benthic community productivity? Or is it inclusive of water column processes as well? While some approaches may be direct metrics of water column + benthic NCP (mass balance or eddy correlation in some cases), others are very clearly metrics of only a single component of NCP. For example, *benthic* chambers explicitly exclude water column production, and are therefore only metrics of *benthic* productivity. Larger benthic chambers which include greater water-column heights may be some combination of water column and benthic NCP. It is therefore hard to understand why all methods were combined, except for 'mass balance'.

168: Again, there *are* studies in this meta-analysis that measured changes in seawater DIC directly.

170-175: The goal of this study was to assess the ability of seagrasses to mitigate coastal OA. This is necessarily a question of carbonate chemistry variability, and as such, I find the DO-based approach used here to be highly suspect. Yes, if PQ and RQ are exactly 1:1, then a conversion of DO-based NCP to DIC-based NCP is appropriate. However, there is no reason to think that either PQ or RQ should be 1:1 in a seagrass meadow where a variety of processes consume/produce DIC (calcification, anaerobic metabolism) irrespective of Oxygen exchange. As such, prior measurements place PQ somewhere between 0.5-2.6 (Turk et al., 2015), and direct comparisons of NCP_DO and NCP_DIC show very weak correlations (Barrón et al., 2006), certainly not 1:1 behavior (Van Dam et al., 2019). As a related factor,

anaerobic metabolic processes in sediments can generate appreciable sediment-water TA fluxes in seagrass meadows. The impact of this on carbonate chemical buffering (thereby OA-amelioration) will depend on the stoichiometry and relative rates of the various anaerobic processes.

While direct measurements of sediment-water TA fluxes as well as PQ/RQ variability are limited for seagrass systems, they are available. Therefore, the approach of Kroeker et al of simply ignoring TA sources/sinks, and assuming a 1:1 stoichiometry for converting DO to DIC is inadequate. In order to address the hypotheses of this study, the authors will need to revisit their meta-analysis and make some effort to 1) address uncertainty in PQ/RQ, and 2) incorporate calcification and anaerobic metabolic TA sources/sinks.

234: As with the previous comment, the "steady state box model" presented here suffers from the same issue described above. By estimating NCP_DIC from NCP_DO, then calculating the effect of this NCP_DIC on pH, the authors are again directly linking DO to pH without considering any biogeochemical processes beyond aerobic respiration and photosynthesis. This is not appropriate, as high rates of denitrification, sulfate reduction, and calcification are known to affect DIC and TA dynamics (thereby pH buffering) in seagrass meadows. Calculating changes in pH from DO alone necessarily ignores anaerobic metabolism and can only result in a positive effect of NCP on pH (as is presented in this box model Figure 6). As such, the simple model presented here does not help to address whether seagrass meadows (not just the seagrass aboveground biomass) can mitigate local OA.

339: "…seagrass meadows in both geographies were net autotrophic…". Yes, these sites were on average net sources of O2, and were therefore net autotrophic with respect to O2. However, net metabolism with respect to C will be lower that O2-based metabolism, due to anaerobic metabolism in sediments (generating DIC and TA). Without accounting for these processes in some way, DO-based metabolism is simply not informative to an assessment of trophic state in the context of OA.

References:

Tokoro, T., Hosokawa, S., Miyoshi, E., Tada, K., Watanabe, K., Montani, S., et al. (2014). Net uptake of atmospheric CO2 by coastal submerged aquatic vegetation. *Global Change Biology*, *20*(6), 1873–1884. https://doi.org/10.1111/gcb.12543

Perez, D. I., Phinn, S. R., Roelfsema, C. M., Shaw, E., Johnston, L., & Iguel, J. (2018). Primary Production and Calcification Rates of Algae-Dominated Reef Flat and Seagrass Communities. *Journal of Geophysical Research: Biogeosciences*, *123*(8), 2362–2375. https://doi.org/10.1029/2017JG004241

Eyre, B. D., Maher, D., Oakes, J. M., Erler, D. V., & Glasby, T. M. (2011). Differences in benthic metabolism, nutrient fluxes, and denitrification in Caulerpa taxifolia communities compared to uninvaded bare sediment and seagrass (Zostera capricorni) habitats. *Limnology and Oceanography*, *56*(5), 1737–1750. https://doi.org/10.4319/lo.2011.56.5.1737

Dollar, S. J., Smith, S. V., Vink, S. M., Obrebski, S., & Hollibaugh, J. T. (1991). Annual cycle of benthic nutrient fluxes in Tomales Bay, California, and contribution of the

benthos to total ecosystem metabolism. *Marine Ecology Progress Series*, *79*(1–2), 115–125. https://doi.org/10.3354/meps079115

Barrón, C., Duarte, C. M., Frankignoulle, M., & Borges, A. V. (2006). Organic carbon metabolism and carbonate dynamics in a Mediterranean seagrass (Posidonia oceanica) meadow. *Estuaries and Coasts*, *29*(3), 417–426. https://doi.org/10.1007/BF02784990

Van Dam, B. R., Lopes, C., Osburn, C. L., & Fourqurean, J. W. (2019). Net heterotrophy and carbonate dissolution in two subtropical seagrass meadows. *Biogeosciences Discussions*, 1–26. https://doi.org/10.5194/bg-2019-191

Ribas-Ribas, M., Hernández-Ayón, J. M., Camacho-Ibar, V. F., Cabello-Pasini, A., Mejia-Trejo, A., Durazo, R., et al. (2011). Effects of upwelling, tides and biological processes on the inorganic carbon system of a coastal lagoon in Baja California. *Estuarine, Coastal and Shelf Science*, *95*(4), 367–376. https://doi.org/10.1016/j.ecss.2011.09.017

Turk, D., Yates, K. K., Esperance, C. L., Melo, N., Ramsewak, D., Dowd, M., et al. (2015). Community metabolism in shallow coral reef and seagrass ecosystems, lower Florida Keys. *Marine Ecology Progress Series*, *538*(Hatcher 1997), 35–52. https://doi.org/10.3354/meps11385

---

## Author Response (AR1)

Response to the editor:

We agree with the reviewers that a meta-analysis of studies measuring carbon metabolic fluxes would be additionally useful, and arguably more targeted in addressing the potential for seagrass metabolism to drive OA amelioration. However, the purpose of the study herein was to review past studies on metabolism, building on previous efforts such as those conducted by Duarte et al. (2010) and Unsworth et al. (2012). Given the high prevalence of $O_2$ studies, we intentionally choose to focus on these measurements as an opportunity to assess broader trends within the literature.

As future research advances and more direct carbon metabolic fluxes measurements are taken, and additional review or meta-analysis on these studies would be a welcome scientific contribution. By re-framing our study to highlight variability in oxygen fluxes, we address the reviewers concerns while still providing a useful synthesis of the many new oxygen metabolism studies released since past review efforts were conducted. In addition to these overarching comments, we address reviewer specific reviewer comments below:

**Reviewer 1:**

Reviewer comment: *136: The authors raise two concerns regarding the existing literature, 1) that no data exist for the N Pacific, and 2) no studies exist using direct DIC measurements. For 1), the authors should see Tokoro et al (2014) and more recent work from the group. Regarding point 2), there are in fact quite a few studies which have relied directly on DIC or DIC+TA measurements to quantify NCP in seagrass meadows. At least one of these studies (Van Dam et al., 2019) was included in this meta-analysis, and it is not clear why the authors chose to omit these and other DIC-based metabolism estimates. Others including DIC measurements include, but are probably not limited to: Perez et al 2018; Eyre et al., 2011; Ribas-Ribas et al., 2011, Dollar et al., 1991; Turk et al., 2015*

Response: We have reframed the paper to focus on seagrass oxygen metabolism, removed the box model, and limited the discussion of OA amelioration.
* * *
Reviewer comment: 143-145: There is another major conceptual flaw here regarding the treatment of data collected using different methods. It is unclear what the authors intend NCP to represent. Is this only a consideration of benthic community productivity? Or is it inclusive of water column processes as well? While some approaches may be direct metrics of water column + benthic NCP (mass balance or eddy correlation in some cases), others are very clearly metrics of only a single component of NCP. For example, benthic chambers explicitly exclude water column production, and are therefore only metrics of benthic productivity. Larger benthic chambers, which include greater water-column heights may be some combination of water column and benthic NCP. It is therefore hard to understand why all methods were combined, except for 'mass balance'.

Response: Our intention was to be transparent about the variability arising from the different

methods used to measure seagrass metabolism. Because of the focus on seagrass in particular, we are by definition trying to characterize benthic NCP. However, as the reviewer noted, some methods are more explicitly focused on the benthic component of NCP (i.e., small chambers) but these methods also have drawbacks (e.g., limited flow that influences the estimates, which has been discussed extensively in the literature). Rather than making a judgment call on which method is better, we decided to show the variability that resulted from the methods (**Fig. S1**). Because the estimates from the mass balance approach were much greater in magnitude and variability than other approaches, even those that also include some aspect of the water column NCP, we removed them from the variability analyses but kept them in the methodological comparison. All other methods could not be easily distinguished from each other visually. We further added lines 210-215 to acknowledge the methodological variation we explored in Fig. S1.
* * *
Reviewer comment: 168: Again, there are studies in this meta-analysis that measured changes in seawater DIC directly.

Response: We removed this statement (and others like it) from the manuscript. See lines 177-180.
* * *
Reviewer comment: 170-175: The goal of this study was to assess the ability of seagrasses to mitigate coastal OA. This is necessarily a question of carbonate chemistry variability, and as such, I find the DO-based approach used here to be highly suspect. Yes, if PQ and RQ are exactly 1:1, then a conversion of DO-based NCP to DIC-based NCP is appropriate. However, there is no reason to think that either PQ or RQ should be 1:1 in a seagrass meadow where a variety of processes consume/produce DIC (calcification, anaerobic metabolism) irrespective of Oxygen exchange. As such, prior measurements place PQ somewhere between 0.5-2.6 (Turk et al., 2015), and direct comparisons of NCP_DO and NCP_DIC show very weak correlations (Barron et al., 2006), certainly not 1:1 behavior (Van Dam et al., 2019). As a related factor, anaerobic metabolic processes in sediments can generate appreciable sediment-water TA fluxes in seagrass meadows. The impact of this on carbonate chemical buffering (thereby OA-amelioration) will depend on the stoichiometry and relative rates of the various anaerobic processes.

Response: We agree with the reviewer that the PQ and RQ are unlikely to be 1:1, and we discuss this in the manuscript. Indeed, we report a wider range of PQ values than 0.5-2.6 in our discussion (Lines 415-416). To deal with this issue, refocused the manuscript on variability in $O_2$ fluxes associated with seagrass metabolism, removed the box model, and limited discussion of potential OA amelioration. We further acknowledge sources of variation in these ratios and their implications for OA amelioration in lines 128-130, 144, 611-614, and 622.

**Reviewer 2:**

Reviewer comment: This study aims to identify drivers of change within submerged vegetated habitats through a meta-analysis of seagrass ecosystem metabolism from studies reporting productivity using oxygen as a proxy for carbon fluxes. While this eventually could be a very useful paper, the manuscript has too many flaws to be accepted in its present condition.

The study has omitted large parts of the literature on seagrass productivity, there are several other, very important studies, many reporting $O_2$ dynamics (especially in the older literature). And why report only oxygen evolution studies? There are also reports from other methods, like e.g. Tokoro et al that measured CO2 directly, or in situ PAM work, like e.g. Gobert et al 2015, and references therein.

Response: We considered the full range of papers on seagrass metabolism, but because our intention was to characterize **the variability** in space and time of seagrass metabolism, we could not find enough studies using other approaches (i.e., measuring $CO_2$ directly) to characterize the variability aspect. We did not include approaches that only measured leaf-scale productivity (i.e., PAM) because we were interested in community metabolism associated with seagrass meadows and their associated community.

Regarding the older literature, it is possible that some studies might have been missed using our search terms while some could have been excluded because they did not meet our criteria for inclusion (which were set to try to increase comparability between studies that we synthesized). We have reported our search terms in the paper, but for further clarity, we have developed an additional comprehensive supplementary table that outlines the papers found using our search terms that were excluded and for what reasons.
* * *
Reviewer comment: The comparison between enclosures and other methods is tricky. At lines 124-126 the authors write: "In contrast, measurements of NCP taken over longer time periods or that incorporate the full 24-hour cycle (full-day NCP) provide insight into the cumulative effect of seagrass on seawater chemistry"
This might not be true (for encolusures) since it has been shown that incubations as long as 24h might yield very low values as the chamber effect will be considerable. Thus these studies might have been severely underestimating productivity. (Olive et al 2016)

Response: We agree with this interpretation. However, we decided that transparency was the best option for characterizing the variability associated with the different methods, and as such, included the plot that showed the differences across the different methods. Visually, we couldn't clearly distinguish between methods except for the mass balance approach. This is evident in Fig. S1 and we acknowledge address methodological variation in lines 210-215.
* * *
Reviewer comment: Overall the authors neglect to account for any changes in seawater carbonate chemistry. Especially calcification (very important in many tropical areas) is forgotten. The formation of CaCO3 have been shown to decrease pH and force CO2 from the water to the

atmosphere. However, some researchers suggest that the net effect of increased productivity and calcification has a positive effect on the overall productivity within the system. This must be discussed properly, a good starting point for such a discussion could be found in e.g. Gattuso et al 1995, Frankignoulle et al 1995 and Mazarrasa et al. 2015.

While we appreciate this suggestion, we accept that there may be too many issues to contend with to adequately translate from oxygen evolution studies to OA amelioration. Thus, we have reframed the manuscript on seagrass metabolism and oxygen evolution and limit discussion of OA amelioration potential. Furthermore, we acknowledge additional drivers of seawater carbonate chemistry change such as those mentioned by this reviewer in lines 128-130, 144, 611-614, and 622.

---

## Author Response (AR2)

Response to the editor:

We have made the minor changes suggested by reviewer 2. Specifically, units have been updated to use the correct notation through all figure captions. Fig. 1 was also reformatted to make points more visible. We additionally note that Fig. 1 was slightly adjusted to reflect the studies that were added or dropped from the analysis when major changes were made to the manuscript in the previous round of edits. No other changes were made to the text of the manuscript, according to reviewer recommendations. The old and new Fig. 1 can be viewed below, for clarity on the exact minor changes made to the figure:

Old Fig. 1:

[Figure]

[Figure]

[Figure]

**Sampling Season**
- Winter
- Spring
- Summer
- Fall
- Multiple Seasons

New Fig. 1:

[Figure]

[Figure]

[Figure]

**Sampling Season**

- ○ Winter
- ○ Spring
- ○ Summer
- ○ Fall
- ○ Multiple Seasons